# Efficient, Breathable, and Compostable Multilayer Air Filter Material Prepared from Plant-Derived Biopolymers

**DOI:** 10.3390/membranes13040380

**Published:** 2023-03-27

**Authors:** Rong Wu, Sneha Shanbhag, P. Ravi Selvaganapathy

**Affiliations:** 1Department of Mechanical Engineering, McMaster University, 1280 Main St W, Hamilton, ON L8S 4L8, Canada; 2School of Biomedical Engineering, McMaster University, 1280 Main St W, Hamilton, ON L8S 4L8, Canada

**Keywords:** compostable face mask, pleats, zein, plant biopolymer, air filter

## Abstract

State-of-art face masks and respirators are fabricated as single-use devices using microfibrous polypropylene fabrics, which are challenging to be collected and recycled at a community scale. Compostable face masks and respirators can offer a viable alternative to reducing their environmental impact. In this work, we have developed a compostable air filter produced by electrospinning a plant-derived protein, zein, on a craft paper-based substrate. The electrospun material is tailored to be humidity tolerant and mechanically durable by crosslinking zein with citric acid. The electrospun material demonstrated a high particle filtration efficiency (PFE) of 91.15% and a high pressure drop (PD) of 191.2 Pa using an aerosol particle diameter of 75 ± 2 nm at a face velocity of 10 cm/s. We deployed a pleated structure to reduce the PD or improve the breathability of the electrospun material without compromising the PFE over short- and long-duration tests. Over a 1 h salt loading test, the PD of a single-layer pleated filter increased from 28.9 to 39.1 Pa, while that of the flat sample increased from 169.3 to 327 Pa. The stacking of pleated layers enhanced the PFE while retaining a low PD; a two-layer stack with a pleat width of 5 mm offers a PFE of 95.4 ± 0.34% and a low PD of 75.2 ± 6.1 Pa.

## 1. Introduction

Air pollutants such as particulate matter, such as black carbon, mineral dust, aerosols, toxic gases, and biological pollutants, are increasingly responsible for a range of respiratory diseases. Advanced air filter materials are being developed to target the highly efficient removal of a range of particulate matter, such as particles with an average diameter of 10, 2.5, or 0.3 microns, while also demonstrating low pressure drop for breathability or reduced energy consumption. While the performance of the filtration media and face masks is of utmost importance to protect human health, the SARS-CoV-2 pandemic has also raised much attention to the disposal and environmental implications of air filter media and face masks, which are usually single-use devices [1]. This in turn has driven much attention to the development of recyclable and biodegradable or compostable air filter materials.

Biodegradable or compostable air filter media and face masks produced from bio-derived or waste-derived biopolymers offer a viable alternative to reducing the environmental impact of air filters and face masks [2]. Air filter media and face masks are often developed as microfibrous nonwovens from melt-blowing synthetic plastics, such as polypropylene. Melt-blown microfibers exposed to an electret process offer high filtration efficiencies for submicron particles while maintaining a low pressure drop [3]. Although polypropylene can be collected and recycled, and reverse supply chains for collecting used filters and face masks at the community scale, separating the filter layers and reprocessing these into functional filters is a challenge. Melt-blowing biopolymers to produce nonwoven filter materials has currently only been feasible with polylactic acid (PLA) [4], a commercially produced synthetic biopolymer that degrades at slow rates under natural conditions [5]. Solution-based processes, such as electrospinning, offer a versatile processing platform to a wide range of natural or synthetic biopolymers that more readily degrade under natural conditions in soil or water [2,6]. However, unlike melt blowing, which produces microfibers, electrospinning produces nanofibrous filter mats that have smaller fiber and pore diameters.

Bio-based air filter materials and face masks need to meet the performance standards for their intended application while being compostable. Several groups have developed electrospun air filter materials using a range of biopolymers, such as polylactic acid, proteins such as zein, cellulose acetate, and chitosan [2]. In previously reported studies on electrospun zein air filter materials, the particle filtration efficiencies range from 93% to 99.99% and the pressure drop ranges between 90 and 175 Pa. However, these studies involved testing the nanofibrous filter materials under lower airflow rates of 2–6 cm/s, and their expected pressure drop would exceed the limits required for breathability when extrapolated to an airflow velocity of 10 cm/s as required by the ASTM standard F3502-21 for barrier face coverings (face masks for bioaerosols) [7,8,9].

Reducing the pressure drop can be achieved via modification to the process parameters to obtain more hierarchical fibrous structures [10,11] or by modifying the structure of the electrospun filter through postprocessing techniques. The former approach requires modification to the electrospinning nozzle to create low and high solution concentration zones, which in turn produce a bimodal distribution in fiber sizes. This approach may be feasible for certain biopolymers and solvent systems. However, the latter approach of postprocessing, such as creating mechanical structures to lower the pressure drop, can be applied to any nanofibrous filter mat. To the best of our knowledge, mechanical structuring has not been explored in the context of electrospun air filter materials or face masks.

In this work, we have developed a compostable air filter material produced by electrospinning a plant-derived protein, zein, on a craft paper-based substrate with high particle filtration efficiency. We assess this filter material for a barrier face covering application or face masks for bioaerosols as provided in the ASTM F3502-21 standard (equipment and aerosol used is identical to that required by NIOSH in 42 CFR Part 84 for testing N95 respirators) [12]. We examine the effect of the concentration of a solution and the behavior of electrospun zein materials in humidified testing conditions. We also examine approaches to enhance the moisture tolerance and robustness of the electrospun zein filter material. We also investigate mechanical structuring to enhance particle filtration efficiency and reduce pressure drop of nanofibrous filter materials. As such, we report a composite nanofibrous filter design that demonstrates high filtration efficiency and low pressure drop over long durations and under high flow rates that have not been demonstrated before.

## 2. Materials and Methods

### 2.1. Materials

A cellulose-based Japanese craft tissue paper, Tengujo (13 gsm), was sourced from a craft paper supplier (Japanese Paper Place, Toronto, ON, Canada). Zein from maize (grade Z3625), citric acid (ACS reagent, 95%), sodium hydroxide (ACS reagent, 97%), and anhydrous ethanol (EtOH, ≥99.9%) were procured from Sigma-Aldrich Inc. Whatman grade 2 filter paper was purchased from Fisher Scientific. A nonwoven melt-blown polypropylene fabric supplied by Ronco Canada was used as the polypropylene-based filtration material for comparison.

### 2.2. Preparation of Electrospinning Solution

Anhydrous ethanol was first diluted to a 70% concentration (*v*/*v*) using deionized water. Uncrosslinked zein solutions of 25%, 30%, and 40% (*w*/*v*) concentrations were prepared in 70% ethanol. Crosslinked zein solutions were prepared using an experimental protocol reported elsewhere [13]. Briefly, a citric acid solution of 9% concentration (*w*/*v*) and a sodium hydroxide solution with a concentration of 0.125 mg/mL were respectively prepared using 70% ethanol as the solvent. The pH of the citric acid solution was adjusted to 4 by a dropwise addition of sodium hydroxide solution to it while monitoring the pH of the solution using a pH meter (model number ST2100-F, Ohaus, Parsippany, NJ, USA). A zein solution of 50% concentration (*w*/*v*) was prepared using the pH 4 citric acid solution, and it was aged for 48 h prior to preparing crosslinked zein solutions with it. The aged 50% zein solution was then diluted using 70% ethanol to concentrations of 25%, 30%, and 40% of crosslinked zein (*w*/*v*). A simplified process describing the preparation of the electrospinning solution is provided in Figure 1a.

### 2.3. Preparation of Electrospinning Substrates

Tengujo paper (13 gsm) was first flattened using a hot iron to remove creases. Flat sheets of Tengujo paper were cut to a size of 8″ × 15″. To prepare folded substrates, we first cut strips from a transparent acetate sheet to a width of 1, 5, 10, and 20 mm. Tengujo paper was pleated in an accordion format using the acetate strip as a guide to prepare 6 sheets each of 1, 5, 10, and 20 mm accordion folds. Further, 3 sheets of each of the pleated widths were stretched out to prepare prepleated flat sheets. A schematic of the pleating process is described in Figure 1b.

### 2.4. Electrospinning Process

Electrospinning of the zein solution was carried out using 6 parallel nozzles with an inner diameter of 0.8 mm with a multinozzle electrospinner/electrosprayer with a drum collector (NE300, Inovenso, Istanbul, Turkey). A schematic of the electrospinning process is provided in Figure 1c. First, we used flat substrates to electrospin uncrosslinked and crosslinked solutions of zein with varying concentrations as described above. All solutions were electrospun on a drum collector at a temperature of 24 °C, relative humidity conditions between 50% and 60%, a distance of 10 cm between the drum and the nozzles, a lateral drum displacement of 5.5 cm (side to side) at a speed of 1 cm/s, and a drum speed of 100 rpm. The electrospinning voltage and solution flow rate, however, were varied with varying concentrations. The viscosity was changed significantly with the increasing of concentration. Both the voltage and the flow rate were adjusted to obtain the optimal conditions for continuous fiber generation. Table 1 summarizes the electrospinning conditions used at various solution concentrations. Each of the various concentrations tested was electrospun at various times of 5, 15, and 25 min, respectively. Finally, to prepare pleated and prepleated electrospun samples, we electrospun 30% crosslinked zein solution onto all pleated tissue papers, which were stretched out over the drum, and taped on the sides of the substrate. The pleated samples then used for further testing were prepared by allowing the folds to be regained by the sample (due to spring back) after electrospinning. Similarly, the prepleated samples were prepared by stretching out and testing the samples in a flat format.

### 2.5. Preconditioning under Humid Conditions

For testing the effect of humidity on the filter efficiency of the filter materials, we exposed a subset of samples of the crosslinked and uncrosslinked electrospun samples to 85 ± 5% RH at 25 °C for 1 h using a temperature–humidity/environmental chamber (ESL-2CW, ESPEC North America Inc., Hudsonville, MI, USA). Following exposure to the humidified environment, the particle filtration efficiency and pressure drop were tested under identical test conditions as before exposure, which are further detailed in the next section. 

### 2.6. Determination of Filtration Efficiency and Pressure Drop

The filtration efficiency of the fabric (test area of 100 cm^2^) was tested using the TSI Certitest automated filter tester 8130A using a sodium chloride aerosol with a count median particle diameter of 75 ± 2 nm (corresponding to 260 nm mass median diameter) at a flow rate of 60 lpm or a face velocity of 10 cm/s as described in the ASTM F3502-21 standard for testing the particle filtration efficiency and pressure drop of the samples. Finally, we also conducted a salt loading test using the above-mentioned airflow conditions where the filtration efficiency and pressure drop profile for unconditioned flat, pleated, and prepleated samples were monitored as the sample was continuously exposed to an aerosol concentration of 20 mg/m^3^ for a duration of 1 h.

### 2.7. Mechanical Folding

Electrospun filter samples based on the selected configurations were exposed to mechanical handling and then characterized again for their performance in terms of filtration efficiency and pressure drop. To mimic mechanical handling during use, we manually folded the electrospun filter repeatedly for at least 4 times, and then unfolded it prior to taking a measurement and studied the degree of detachment of the fibers after mechanical handling.

### 2.8. Biodegradability

Biodegradability tests were conducted in an aerobic soil compost mixture and were carried out at 58 ± 2 °C (ASTM 5338) over a period of 28 days. Briefly, squares (5 cm × 5 cm) of Tengujo tissue paper effectively weighing 0.72 g were distributed in 240 g of moist compost. Similarly, tissue paper squares with a crosslinked electrospun zein coating with a total weight of 0.82 g were distributed between layers of 240 g of moist compost. Finally, similar-size squares of Whatman grade 2 filter paper weighing 1 g were used as a cellulosic standard material and were distributed between several layers of 240 g of moist compost. In each case, consecutive layers of compost and few squares of the respective paper were spread out in a sealable glass container. A few drops of water were added to ensure that the contents remained moist. The container was then left open with a gap to aerate the sample and placed in an oven maintained at 58 ± 2 °C. For each sample type, we had 2 containers containing the respective sample. The disintegration of the samples was observed at the end of every week by extracting 2 squares of each paper/filter type. Soil was brushed off gently from the extracted paper samples, and they were gently washed with water and dried for 2 h in an oven at 90 °C. Photographs of the clean samples were taken to demonstrate the degradation of the samples under composting conditions.

### 2.9. Surface Characterization

We collected scanning electron microscopy (SEM) images to investigate the surface and fiber morphology of the electrospun samples using a TESCAN VEGA -II LSU SEM, Cambridge, UK. Sections (0.5 cm × 0.5 cm) of electrospun fabric samples around the central area were coated with a 5–10 nm layer of gold using a sputter coater to improve the surface conductivity, and SEM images were collected at 20 kV in the secondary electron imaging mode. SEM images were collected for electrospun samples for each concentration condition for crosslinked and uncrosslinked samples. Additionally, SEM images were also collected after before and after exposing the electrospun samples to high humidity, mechanical handling, and salt loading tests.

## 3. Results and Discussion

### 3.1. Morphology of the Composite Zein Filter and Filtration Performance

We characterized the morphologies of electrospun zein fibers prepared from 25%, 30%, and 40% (*w*/*v*) of zein solutions respectively using SEM. The distribution of the fiber diameter of zein fibers was obtained from the SEM images. As shown in Figure 2, the fiber diameters and structures are significantly different for various concentrations. It can be seen from Figure 2a that the fibers from a 25% concentration of zein solution consist of ultrafine fibers with a ‘beads on string’ structure, whereas bead-free fibers were obtained from a 30% and 40% concentration of zein solution. By comparison between Figure 2b and c, ribbonlike, wide, and flat zein fibrous structures can be found in the SEM of a 40% solution concentration caused by the high volatility of ethanol that produces a skin on the jets of the zein solution, as has been described in previous studies [14,15]. Fiber diameter distributions for the various concentrations studied are illustrated in Figure 2d–f. The average fiber diameters for the 25%, 30%, and 40% solutions are 0.15, 0.30, and 0.34 μm, respectively. For the formation of fibers, a minimum concentration of solution needs to be satisfied. In the previous studies, a mixture of beads and fibers is obtained at a low solution concentration. With increasing solution concentration, the shape of the beads on the fibers changes from spherical to spindle-like, and finally, due to the higher viscosity resistance of the solution, uniform fibers with increased diameters are formed [16]. However, the formation of continuous fibers is unstable at high concentrations because of the difficulty to maintain the flow of the viscous solution at the needle tip, which would also lead to the formation of larger fibers [17]. From SEM images and the fiber diameter distribution, 30% of zein solution produced the most stable and continuous fiber for a composite filter based on the balance of solution viscosity and electrospinning parameters. Our tested concentration range is similar to previous reports of zein-based air filters [8,9,11].

To evaluate the structural advantages of fibers prepared from various solution concentrations, we investigated the air filtration performance of composite zein filters with different durations of electrospinning time. Both particulate filtration efficiency (PFE) and pressure drop (PD) are characterized as shown in Figure 2g–i. Notably, the PFE and PD of the Tengujo tissue paper substrate are 0.77 ± 0.13% and 2.1 ± 0.4 Pa, indicating that the increase in PFE and PD should be mainly attributed to electrospun fibers. An SEM image of the substrate is shown in Appendix A. At the lowest spinning time of 5 min, the filter produced from 40% zein presents the lowest PD (18.2 ± 0.4 Pa) but also the lowest PFE (16.97 ± 0.41%) with a flow rate at 85 lpm. In contrast, the thinner nanofiber filter from 30% zein offers a higher PFE (30.73 ± 0.52%), but also offers a higher airflow resistance, rendering a larger pressure drop (23.1 ± 0.6 Pa). The 25% zein solution produced the finest fibers, with a bead on the string structure, but offers a slightly lower PFE (25.89 ± 0.77%) and PD (19.2 ± 0.4 Pa) probably because of the incomplete or nonuniform surface coverage for the 5 min sample, resulting from a smaller Taylor cone. Interestingly, the filtration efficiency of filters prepared with 30% zein spun for 25 min was 91.15%, which is the best among all the three concentrations, but its pressure drop was as high as 192.2 Pa. The fiber sizes of the 30% zein solution appear to favor the interception of the salt aerosol particles (0.075 μm count median particle size) used in PFE tests, resulting in the high removal efficiency [9]. Due to its higher filtration efficiency and absence of beaded structures, the 30% zein solution was selected in following experiments and further analysis.

### 3.2. Effect of Crosslinking

#### 3.2.1. Morphology and Air Filtration Performance

The electrospun fibers of crosslinked solution and uncrosslinked solution were characterized under SEM as shown in Figure 3a,b under identical magnification. Both conditions produce a smooth and continuous fiber by electrospinning effectively. The average diameter of a crosslinked fiber is higher than that of an uncrosslinked fiber, which is 0.30 μm and 0.16 μm, respectively, in Figure 3e,f. As reported previously, 9% citric acid was used as a crosslinker since it shows remarkable improvement of multiple properties as a crosslinker for high-concentration protein solution from previous studies [13]. During the crosslinking process, the reaction between an amine group of zein and the carboxylic acid group in citric acid results in amido linkages, which might lead to a slightly larger fiber diameter compared with an uncrosslinked fiber [18]. The air filtration performance was also tested for both the crosslinked and uncrosslinked 25 min electrospun filter. The uncrosslinked filter shows a similar PFE at 91.46 ± 2.31% with the crosslinked filter at 91.14 ± 1.24%, but a slightly higher PD at 189.2 ± 18.4 Pa compared with 181.3 ± 21.9 Pa for the crosslinked filter. The lower fiber density and decrease of the air resistance may be related to the increasing diameter of fibers of the crosslinked filter [19]. The performance of both crosslinked and uncrosslinked zein is similar as they both possess various functional groups that interact with a range of particulate pollutants and enable the purification of the air [14,20,21]. Although uncrosslinked zein offers excellent performance of air filtration, the crosslinking process improved other properties, which will be discussed in the next sections.

#### 3.2.2. Effect of Humidity on Filtration Performance

The cross-linking with citric acid also affected the surface morphology of zein-based nanofibers under high relative humidity (RH 85%). The stability of the filter under high humidity was evaluated by SEM and air filtration performance for both PFE and PD with 0, 30, and 60 min of conditioning under high humidity conditions in Figure 3c,d. The fibrous structure of the uncrosslinked filter was destroyed with significant swelling and fusion of fibers after 30 min, and cracks were visible over the entire area (Appendix A), indicating that the pure zein nanofiber materials had poor moisture resistance under high humidity for long time duration. The filtration performance decreased to 79.24 ± 5.35% after 30 min and 59.75 ± 6.22% after 60 min exposure. The PD showed a similar trend and dropped to 136.4 ± 19.4 Pa at 30 min and 92.9 ± 15.3 Pa at 60 min. The dramatic drop of PFE and PD is mainly attributed to the crack formation in the filter surface, the remaining function as air filter being provided by some portions of the swollen electrospun fibers. Although previous work showed good stability of the uncrosslinked zein fiber under humidified conditions (10 s exposure during testing only) [22], our results demonstrate that the uncrosslinked zein layer cannot resist high relative humidity as would be experienced in the enclosed region of the face mask when a person is wearing it. In contrast, crosslinked zein retained the fibrous shape, as illustrated in Figure 3d, which indicates that crosslinking with citric acid improved the stability of zein nanofibers under a humidified environment. The amido linkages formed during crosslinking improved the moisture tolerance of the crosslinked samples [13].

As shown in Figure 4a,b, there was a slight reduction of PFE to 89.42 ± 2.76% at 30 min and 87.85 ± 4.02% after 60 min of exposure, and PD increased to 183.6 ± 19.0 Pa and 184.2 ± 17.8 Pa, respectively. The minor loss in performance is expected to be a result of the slight swelling of the zein fiber even after the crosslinking process. However, the significantly lower degree of swelling resulting from the crosslinking process may have resulted in preventing the formation of cracks on the crosslinked filter surface [13]. As such, a filter material prepared with crosslinked zein is amenable for air filtration applications, where exposure to higher humidity conditions is expected. The degree of crosslinking of zein solution can be optimized to ensure long-duration performance under humid conditions.

#### 3.2.3. Mechanical Handling Stability

The poor mechanical performance of nanofibrous electrospun filter materials limits their practical application as an air filter. To validate the robust mechanical handling of the zein fiber fabricated in this work, both the uncrosslinked and crosslinked zein filters have been tested by folding four to five times per test manually, and the results of the PFE and PD are shown in Figure 4c,d. The PFE of the crosslinked fabric remained stable, indicating the robustness and durability of the crosslinked filter material. In contrast, the PFE of the uncrosslinked zein layer decreased continuously with an increasing number of folding. The value of the PD shows a similar tendency, with minor changes in the crosslinked zein, and a dramatic drop of the uncrosslinked fabric. The slight increase observed in the pressure drop values of the crosslinked sample is expected to have been caused by the accumulation of ultrafine particles with repeated testing on the same sample. We expect two factors to influence the performance of the filter material after mechanical handling; the first is bonding or attachment between the zein layer and the paper substrate, and the second is the mechanical property of zein fibers. We observed that the crosslinked fiber did not detach from the paper substrate easily after exposing to repeated folding and testing, suggesting that the crosslinking process formed linkages not only between the zein fibers but also with the paper substrate, thereby enhancing its mechanical stability. Additionally, the formation of an amido linkage is also expected to improve the tensile strength of the crosslinked zein fibers [13,23].

### 3.3. Air Filtration Performance of Pleated Electrospun Filter

To improve filtration performance, we applied a pleated filter format as previously investigated in cartridge air filters [24] to face masks or respirators. We investigated the effect of pleating by preparing pleated electrospun filters with varying pleat widths of 1, 5, 10, and 20 mm using crosslinked 30% zein filter material. The filtration efficiency and pressure drop observed for these are illustrated in Appendix A. The PFE is 88.48%, 84.26%, 82.76%, and 78.43%, and the PD is 138.9, 34.1, 36.7, and 36.1 Pa, respectively, for 1, 5, 10, and 20 mm pleat width filters. A size of folds smaller than 5 mm is not favorable for lowering air resistance likely because of a relatively small increase in surface area and a smaller effect on the flow profile of the aerosol. A pleat size with 5 mm was found to provide the best performance and utilized for further investigation. Although the samples with 10 and 20 mm pleats offered improved breathability over the flat sample, their PD values were still slightly higher than those of the 5 mm pleats. Preventing the larger pleats from collapsing onto one another and retaining the desired triangle pleating shape during testing was challenging, and may have influenced our observed results for larger size pleats. Therefore, the 5 mm pleat size was used for investigating the effect of stacking layers of pleated filters.

We also found that the PFE of pleated samples was slightly lower than that of their flat counterparts by 6.9% from 91.15% to 84.26% for the 5 mm pleat width sample. We expect that the pleat/crease lines influence the flatness of the paper substrate while electrospinning, leading to a gradation of fiber coverage along the pleat/crease line. Additionally, the PD significantly decreased by 82.3% from 192.2 Pa of the flat sample to 34.1 Pa of the pleated sample. The PD across a pleated filter material can be attributed to the PD caused by the filter fibers and that offered by the structure geometry [25]. By incorporating the pleated structure, the total filtration surface area increases, and the face velocity effectively decreases. The pleated sample area is 2.4 times higher than that of the flat sample, which partially contributes to lowering the PD. The gradation of the fiber coverage between the body and crease lines (trough and peak of accordion fold) on the pleated sample and the airflow profile (Appendix A) may also play a role in lowering the resistance against aerosol flow and pressure drop. We expect that the aerosol flow tends to penetrate the pleats through the weak or thin wall of filter materials where the nanofiber coverage and airflow resistance are low [26]. Our results suggest that pleats are advantageous to lower the pressure drop from the dense nanofibrous filter.

Despite the high removal efficiency at 91.15% achieved by electrospun zein filter materials, the PD is still high at 191.2 Pa. The excellent breathability from the pleated zein sample indicates that introducing folded structures is favorable to lower the overall airflow resistance while maintaining a high PFE. In addition, face velocity was considered when different filters were evaluated since the PFE and PD would be both influenced at a higher face velocity [27]. Both parameters were tested under four different face velocities from 2 to 14 cm/s, as shown in Appendix A. The filtration efficiency remained around 95% when the face velocity increased, indicating that the PFE remained consistent over a wide range of airflow velocity to support daily usage. Meanwhile, the PD of the tested composite filter linearly increased with the face velocity, from 18.4 to 106.2 Pa.

### 3.4. Stacking of Electrospun Zein Filters

The improvement in the filtration efficiency and airflow resistance of the multilayered composite filter with flat, prepleated, and pleated configuration is summarized in Figure 5a. All three configurations of zein filter have excellent filtration function with over 99% for multilayer combinations. For flat and prepleated structures, the PFE is even over 99.9%, with an extremely high PD at 1219.1 and 591.0 Pa for the five-layer stack. Although the PD linearly increases with the number of layers for all three structures, we observed that the pleated stacks have the lowest PD for any number of layers. The increment of the PFE is smallest, and its rate of increase with layers is lowest in pleated samples due to the high surface area. We observed that despite the stacking of layers, the PD of the pleated stacks is only 1/5 that of the flat filter stacks. Specifically, a two-layer pleated stack offers a PFE of 95.4% and a PD of under 100 Pa, while a two-layer flat stack offers a PFE exceeding 99% and a PD of nearly 500 Pa. Our findings indicate that the pleated structure is beneficial in lowering the PD of the electrospun samples even in a multilayer stack. Further, this composite pleated stack meets the requirements of a level 1 barrier face covering (20% PFE, 149 Pa PD) and may be suitable for creating an N95 respirator.

To compare the filtration performance of the filter material in this work with previous studies, the quality factor (QF, unit: Pa^−1^) is calculated based on the equation below:$$QF=\frac{-ln(1-\eta )}{\Delta P}$$
where η is the particle removal efficiency, and ΔP is the pressure drop of the air filter.

QF indicates the overall performance of filtration based on the PFE and PD. A comparison of filtration efficiency with other air filters is summarized in Figure 6a. Although the testing parameters, such as particle size and flow rate, varied in all those studies, our testing environment followed the ASTM F3502-21 standard, which is more conservative. Even at an extremely high face velocity of 14 cm/s (numbering in the x-axis data labels indicates face velocity in cm/s), the QF of this work is higher than most of other previous studies at comparable flow rates. This further exhibits the comprehensive air filtration performance with high PFE and low PD producing by nanofibers and folding structures.

### 3.5. Extended Filtration Performance of Composite Zein Filter

To evaluate the filtration performance and the airflow resistance of the pleated air filter made of zein with different structures over a longer duration of time, we conducted a 1 h loading test for removing ultrafine salt particles. As shown in Figure 6a, the PFE of most of the filters are over 99% after long time exposure except the pleated one-layer zein filter. The PFE of the single-layer pleated filter increased by 8.31% from 87.11% after 1 h of testing, whereas that of the flat sample increased by 7.04% from 92.86% and the prepleated by 15.30% from 83.69%. Interestingly, the prefolded sample shows the highest increase, likely due to particles being trapped and attracted to open areas for aggregation, thus reducing the effect of inhomogeneous coverage. After the large open areas on prepleated structures are filled with fine particles, mechanical filtration dominates the loading process, and the efficiency continues to increase further. The advantage of the folded structure is more obvious in the pressure drop performance. After a 1 h loading test, the PD increased from 28.9 to 39.1 Pa for the pleated sample, from 169.3 to 327 Pa for the flat sample, and from 103.2 to 334.7 Pa for the prefolded sample. The advantage of the pleating structure is further evident in the PD profiles for the prepleated and flat samples, which, despite their initial differences, start to perform similarly as they obtain loading and become saturated with salt particles, and offer a higher resistance to airflow. Pleated structures, therefore, help lower the magnitude and the rate of increase in pressure drop during particle loading. This reflects that the combination of a nanofiber layer and pleating structures helps to achieve the improvement of ultrafine particle efficiency without excessive increase in airflow resistance in both short- and long-term use. Further, the PD increments of 10.2 and 24.7 Pa for single- and two-layer pleated stacks over the 1 h loading test were almost linear with the number of layers. Additionally, the 1 h loading test was performed on a commercial polypropylene (PP) filter layer to compare between the flat and pleated structure, and the results are presented in Figure 6b. Pleats on the PP filter showed a significant improvement for both PFE and PD. The resistance was reduced by 78.6%, while the PFE was even up to 99.67% after loading tests. Therefore, the combination of nano-sized fibers and millimeter-scaled pleats is not dependent on the material or fiber processing method and could be universally applied to other polymer filters used in face masks.

To further interpret the effect of a pleated filter structure, the distribution of particles after long-term testing was also investigated by SEM images captured from different filters. From the SEM results of pleated filter on three different positions, peak, side wall, and trough (Figure 6c–h), the particles tended to predominantly accumulate on the peaks and troughs for both the top and bottom of the top layer filters. With a large height-to-width ratio of the pleat structure and a small pleat pitch, the dust particles are easily deposited in the trough of the pleat and reduce the effective filtration area of the filter media, which will contribute to the increase in pressure drop [32]. Based on SEM images, we observed slightly more particles on the peak area compared with the trough area, which is different from micron-size particles, which deposit largely in the peak angle or trench region [33]. First, unlike larger microscopic particles that have considerable inertia and follow the centerline of the pleat, the ultrafine particles utilized in our study likely follow the streamlines and can settle at the peak and throughout the pleat length [26]. Second, triangular-shaped pleats have been shown to have less pronounced settling in the trench due to the milder turn at the entrance of the airflow at the pleat inlet. This geometry prevents the formation of a ‘dead zone’ to some extent with a relatively uniform distribution of airflow for the pleat peak and bottom [26,34]. Further, based on the SEM images, the side walls of the pleats capture only a small number of particles, leaving a considerable filtration area available even after long-term testing, indicating that the pleated zein filter is long-lasting with stable air filtration performance. This likely also contributes to the low resistance of pleated filters during long-term tests. As indicated by the SEM images, the upper filter layer performs most of the particle filtration, and the bottom layer has an almost clean and available filtration area even after 1 h of continuous testing. In contrast, a considerable amount of particle deposition and blockage of an open surface area on flat and prepleated air filters (Appendix A) results in dramatically high air resistance.

### 3.6. Biodegradability

The zein composite filter with the desired filtration efficiency in this study is made entirely of biodegradable plant-derived materials including the paper substrate. Composted soil degradation was selected to verify the biodegradability/compostability of the zein air filter. Both zein filter and commercial cellulose filter paper were buried in the composting condition at 58 ± 2 °C according to the ASTM standard D5338 [35]. It has been found that the zein filter was fully decomposed in composting soil within 4 weeks by microorganisms, as shown in Figure 7, while a small amount of residue remained in the cellulose filter material. Therefore, the zein composite filter has excellent biodegradability and provides an environmentally friendly solution to the disposal of waste filters.

## 4. Conclusions

In this work, we have developed a compostable air filter material produced by electrospinning a plant-derived protein, zein, on a craft paper-based substrate. The electrospun material is tailored to be humidity tolerant by crosslinking zein with citric acid, and this in turn also enhances the mechanical durability of the filter material. The electrospun material inherently demonstrates a high PFE and a nominally high PD. We deployed a pleated structure to reduce the PD of the electrospun material and demonstrate that stacking several layers of pleated electrospun sheets can offer a simple solution in developing a composite nanofibrous filter with a high PFE of 95.4 ± 0.34% and a good breathability of 75.2 ± 6.1 Pa for a two-layer stack. This composite pleated stack meets the requirements of a level 1 barrier face covering (20% PFE, 149 Pa PD) and may be suitable for creating an N95 respirator. Finally, long-duration testing of this composite filter indicates that the PFE of the filter increases over the test duration. However, the increase in PD of pleated samples and stacks is far lower than of their flat or prepleated counterparts by more than 90%. In effect, pleating as an approach to reduce the PD without significantly compromising the PFE was demonstrated to be a strategy that can be applied to other filter materials.

## Figures and Tables

**Figure 1 membranes-13-00380-f001:**
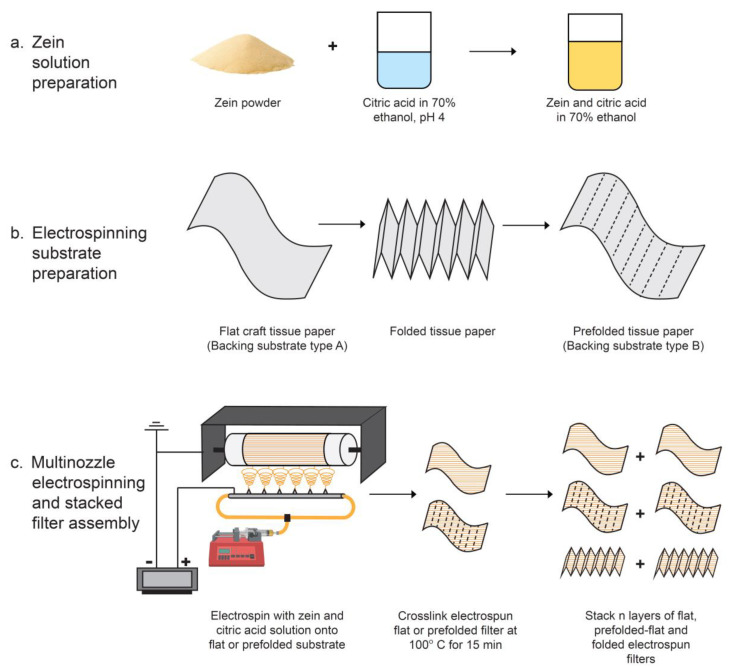
Schematic of the process for preparing compostable multilayer electrospun air filter: (**a**) zein solution preparation, (**b**) electrospinning substrate preparation, and (**c**) multinozzle electrospinning and stacked filter assembly.

**Figure 2 membranes-13-00380-f002:**
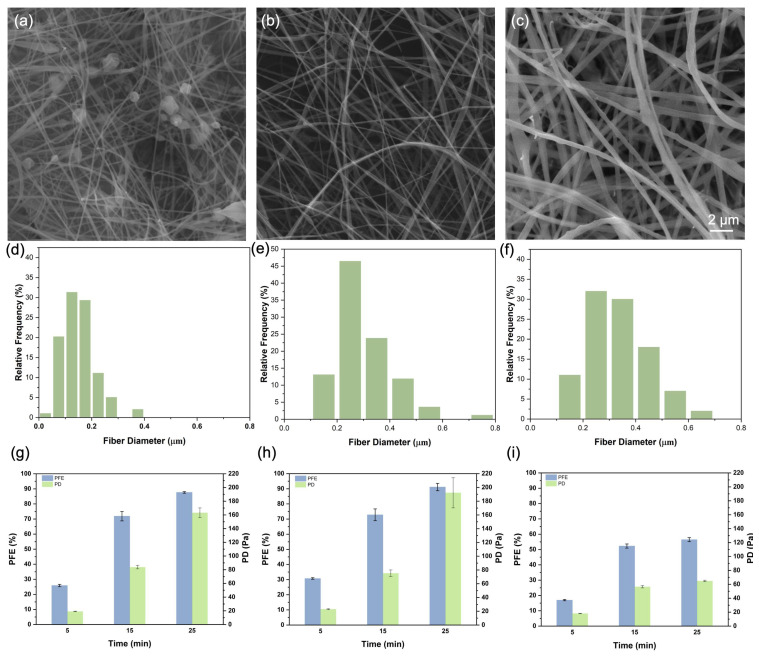
SEM of (**a**) 25%, (**b**) 30%, and (**c**) 40% fibers under a magnification of 8000× and fiber diameters of (**d**) 25%, (**e**) 30%, and (**f**) 40% of crosslinked zein solution; air filtration including PFE and PD performance of (**g**) 25%, (**h**) 30%, and (**i**) 40% with three different electrospinning times from 5 to 25 min.

**Figure 3 membranes-13-00380-f003:**
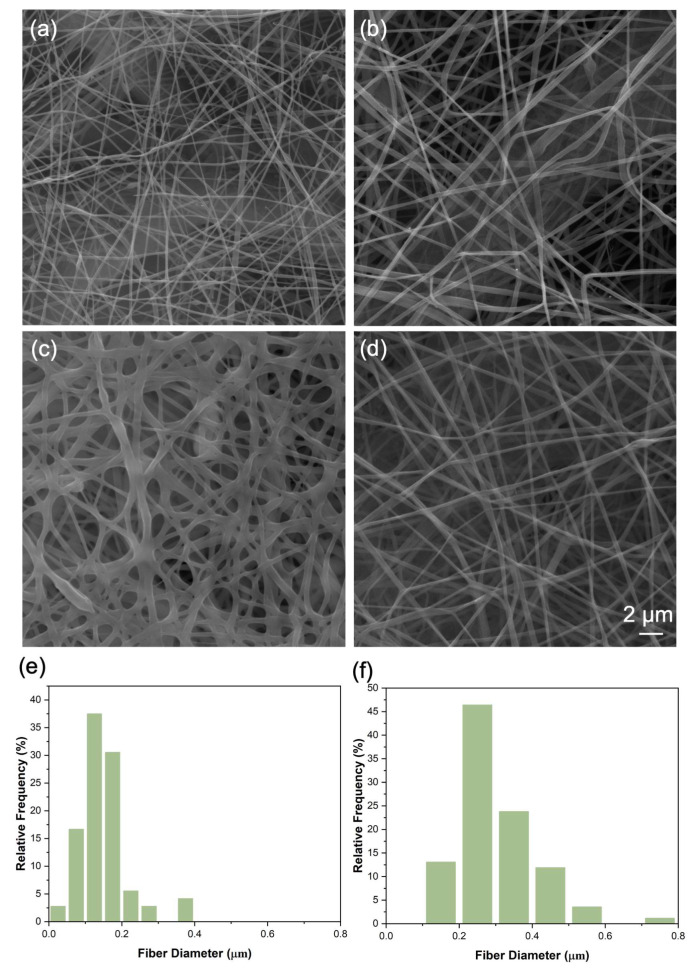
(**a**) SEM uncrosslinked zein fiber, (**b**) SEM of crosslinked zein fibers, (**c**) SEM of uncrosslinked zein fiber exposed to moisture for 60 min, (**d**) SEM of crosslinked zein fiber exposed to moisture for 60 min exposed to moisture (magnification 8000× applied under SEM), (**e**) uncrosslinked zein fiber diameter distribution, (**f**) crosslinked zein fiber diameter distribution.

**Figure 4 membranes-13-00380-f004:**
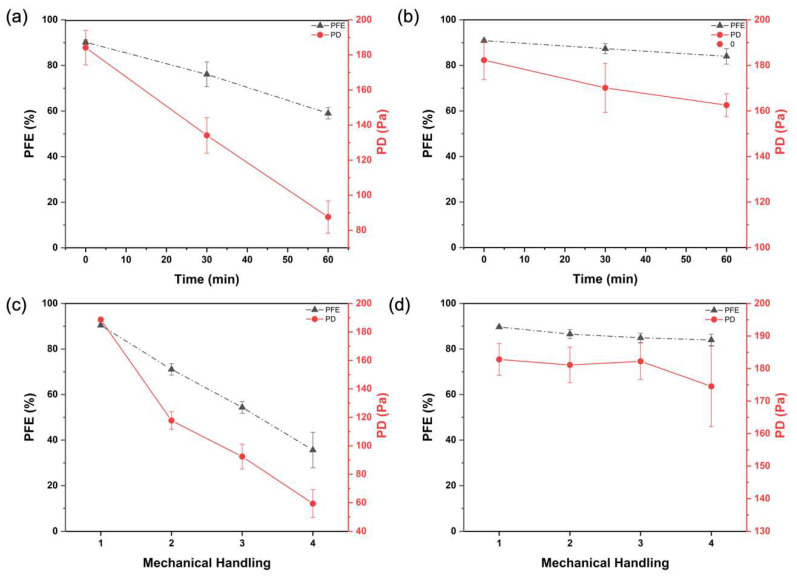
PFE and PD of (**a**) uncrosslinked zein filter with different time of conditioning, (**b**) crosslinked zein filter with different time of conditioning, (**c**) uncrosslinked, and (**d**) crosslinked zein filter after mechanical handling.

**Figure 5 membranes-13-00380-f005:**
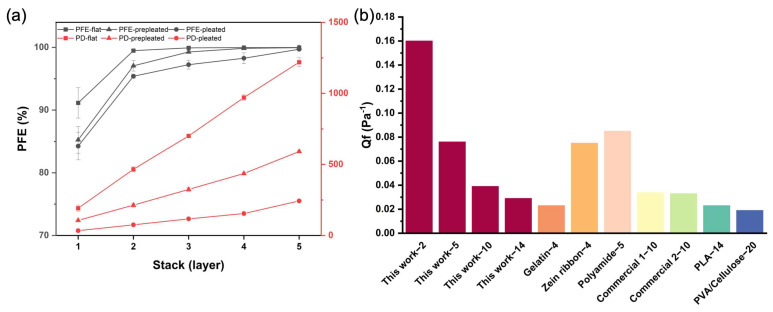
(**a**) Stacking study of zein filter with flat, prepleated, and pleated surface. (**b**) Comparison of quality factors (Qf) between this work at different flow rates and previous studies of synthetic polymer-based and natural polymer-based air filters [8,28,29,30,31]. The digit after the hyphen on data labels indicates the flow rate in cm/s used in these studies.

**Figure 6 membranes-13-00380-f006:**
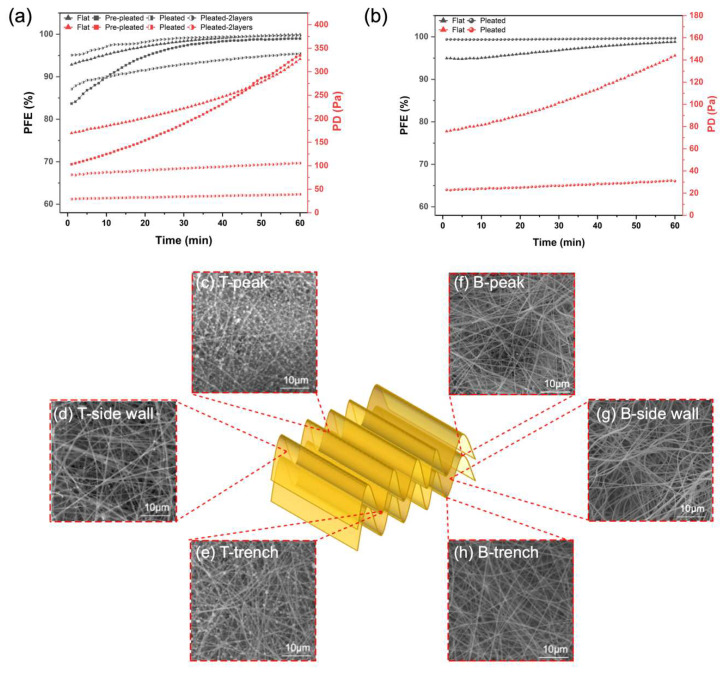
(**a**) Air filtration performance of 60 min loading test of flat, prepleated, and pleated zein filter; (**b**) air filtration performance of 60 min loading test of flat and pleated commercial polypropylene mask filter; schematic and SEM image of (**c**) peak, (**d**) side wall, and (**e**) trench of top pleated filter; (**f**) peak, (**g**) side wall, and (**h**) trench of bottom pleated filter.

**Figure 7 membranes-13-00380-f007:**
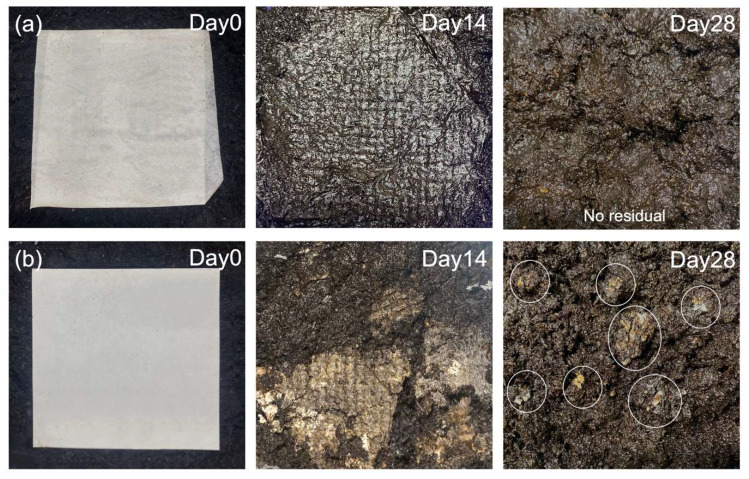
Biodegradability of the developed composite zein filter: (**a**) zein filter degradation within 28 days; (**b**) cellulose filter paper degradation within 28 days.

**Table 1 membranes-13-00380-t001:** Electrospinning conditions for varying zein concentrations.

	Electrospinning Conditions								
Concentration (%, *w*/*v*)	25%			30%			40%		
Substrate type	Flat			Flat			Flat		
Voltage (kV)	21 kV			23 kV			25 kV		
Flow rate (mL/h)	6			8			10		
Number of nozzles	6			6			6		
Relative humidity (%)	50–60			50–60			50–60		
Temperature (°C)	24			24			24		
Distance (cm)	10			10			10		
Homogeneity/lateral distance (cm)	5.5			5.5			5.5		
Homogeneity speed (cm/s)	1			1			1		
Drum speed (rpm)	100			100			100		
Spinning time (min)	5	15	25	5	15	25	5	15	25

## Data Availability

Not applicable.

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
