# Peer review of "Efficient, Breathable, and Compostable Multilayer Air Filter Material Prepared from Plant-Derived Biopolymers"

_membranes, 2023, doi:10.3390/membranes13040380_

Round 1

Reviewer 1 Report

The paper theme is relevant, current and due to covid-19 it has drawn attention.

In general, the work is very good, only the details that will be described below are missing.

Despite the manuscript text being clear, some important information was missing to give more credibility to the presented results.

For example:

- The title, "Efficient, Breathable, and Compostable, Multi-Layer Air Filter Prepared from Plant Derived Biopolymers", putting "Air Filter" is very generic, in fact what was prepared was an air filter material (filter medium ).

- In the summary, .... the aerosol diameter and velocity were not specified to obtain this efficiency and pressure drop.

- In the introduction, better specify the filter medium application that is being developed in this work, important to improve the discussion of the results. In this case, make clear the differences in standards for testing air filters (for their various applications), surgical masks, respirators (N95, PFF), among others, because there are standards, with several operational parameters (speed, flow, different aerosol sizes, among others) to achieve the desired efficiency and pressure drop for each application. This should be very clear in the text, because the authors cite air filters and masks throughout the text.

Therefore, the question remains: the tests to characterize the filtering media developed in this study were carried out with a view to which application(s)?

- Make this very clear in the objectives.

-- In materials and methods, show whether the air filtration tests were carried out according to the standards adopted for air filters, or masks, respirators, etc., and highlight the differences. This is because, for respirators such as N95, the particle size for testing would have to be 0.3 microns. For each application, the parameters: surface velocity, mass flow rate, particle size, vary.

In the electrospinning description, mention the needle diameter used in electrospinning.

For the SEM analysis, what is the size of the samples taken from the filter? how many samples were taken? What is the position (central)?.

What program was used to determine fiber diameter?

- Results and discussion: If tests were performed for air filters, surgical masks, or respirators, among others, specify in the results text that the filtration efficiency and pressure drop refer to a certain application.

Avoid writing optimal, small, high, efficiency, pressure drop values, among others, cite the values obtained in the results and compare these values with those in the literature, or the referred standards.

This is because the value of the pressure drop for an N95 respirator is around 340Pa, this value for a HEPA filter would be great, however, for a surgical mask it would be terrible, high, ideal values would be below 50 Pa.

The Quality factor gives an idea if the filter is good or bad, but it cannot be based on this value to highlight whether a filter is good or bad, there are other factors that must be taken into account, such as: all authors or experiments were performed with the same filtration velocity, mass flow rate, particle size, particle size distribution, etc?

Author Response

Comment #1: - The title, "Efficient, Breathable, and Compostable, Multi-Layer Air Filter Prepared from Plant Derived Biopolymers", putting "Air Filter" is very generic, in fact what was prepared was an air filter material (filter medium).

Response: The authors thank the reviewer for their careful review and feedback. We have modified the title of the manuscript to "Efficient, breathable, and compostable, multi-layer air filter material prepared from plant derived biopolymers"

Comment #2: - In the summary, .... the aerosol diameter and velocity were not specified to obtain this efficiency and pressure drop.

Response: Thank you for the suggestion. The aerosol diameter and face velocity used for testing the material have now been specified and added in the abstract on line 17 to clarify the testing conditions.

Comment #3: - In the introduction, better specify the filter medium application that is being developed in this work, important to improve the discussion of the results. In this case, make clear the differences in standards for testing air filters (for their various applications), surgical masks, respirators (N95, PFF), among others, because there are standards, with several operational parameters (speed, flow, different aerosol sizes, among others) to achieve the desired efficiency and pressure drop for each application. This should be very clear in the text, because the authors cite air filters and masks throughout the text.

Therefore, the question remains: the tests to characterize the filtering media developed in this study were carried out with a view to which application(s)?

- Make this very clear in the objectives.

Response: We thank the reviewer for this critical feedback. We have revised the manuscript to clarify that this work is presently focused on air filter materials that can be used in biodegradable barrier face coverings and the testing is conducted per the ASTM F3502 standard as indicated on lines 62 and 76-78 of the introduction section in our manuscript. We expect that future work could focus on extending the use of this biodegradable filtration material to surgical face masks or other air filtration applications.

Comment #4: - In materials and methods, show whether the air filtration tests were carried out according to the standards adopted for air filters, or masks, respirators, etc., and highlight the differences. This is because, for respirators such as N95, the particle size for testing would have to be 0.3 microns. For each application, the parameters: surface velocity, mass flow rate, particle size, vary.

Response: Thank you for your thoughtful comment. The test parameters of proposed filtration material were set based on ASTM F3502 standard for barrier face covering and the details including the mass flow rate, surface velocity. They have been included in section 2.6  and used throughout the manuscript expect in cases where we intentionally changed one of the test parameters to study the effect. The ASTM F3502 standard has a lower filtration efficiency and pressure drop cut-off than those required to meet the NIOSH requirements 42 CFR Part 84 for N95 respirators. However, the testing equipment and protocol used for both these standards are identical and require a sodium chloride aerosol with a count median particle diameter of 75 ± 2 nm (~260 nm mass median diameter).

Comment #5:  In the electrospinning description, mention the needle diameter used in electrospinning.

Response: Our electrospinning equipment uses nozzles instead of needles. The nozzle diameter of 0.8 mm has been added and highlighted in the manuscript on line 119.

Comment #6:  For the SEM analysis, what is the size of the samples taken from the filter? how many samples were taken? What is the position (central)?

Response: That information was added and highlighted in the manuscript on line 185. We collected samples from the central area to show more representative data. For the characterization of morphology and diameter distribution, 3 sections from 3 various electrospun samples were used to count and plot.

Comment #7: - Results and discussion: If tests were performed for air filters, surgical masks, or respirators, among others, specify in the results text that the filtration efficiency and pressure drop refer to a certain application.

Response: We thank the reviewer for their suggestion. We have revised the manuscript to clarify that the filter material developed was tested for barrier face covering application.

Comment #8: Avoid writing optimal, small, high, efficiency, pressure drop values, among others, cite the values obtained in the results and compare these values with those in the literature, or the referred standards. This is because the value of the pressure drop for an N95 respirator is around 340Pa, this value for a HEPA filter would be great, however, for a surgical mask it would be terrible, high, ideal values would be below 50 Pa.

Response: We thank the reviewer for their suggestion. We have replaced the unclear description in the manuscript with the values and results.

Comment #9: The Quality factor gives an idea if the filter is good or bad, but it cannot be based on this value to highlight whether a filter is good or bad, there are other factors that must be taken into account, such as: all authors or experiments were performed with the same filtration velocity, mass flow rate, particle size, particle size distribution, etc?

Response: We thank the reviewer for their thoughtful comment. We have provided the testing details regarding face velocity, flow rate, particle size in the section 2.6 for the experiments and all parameters were selected based on the ASTM standard F3502. For the study of face velocity (2-14 cm/s), we used various flow rates and have specified those in the manuscript. For the comparison with other studies, even though factors like particle size, flow rate are varied, we have used more conservative settings according to the standard which means if we use their settings, better results are expected. The face velocity for each study (from literature) has been indicated in the Quality factor plot for comparison.

Reviewer 2 Report

The authors developed a biodegradable membrane for air filtration. This work is recommended for publication in Membranes after the following comments are clarified.

1.     The description of the electrospinning process needs to be further refined. In particular, how the pleated substrates were fixed on the drum to maintain its shape.

2.     The microstructure, filtration efficiency and pressure drop of craft tissue paper without electrospinning should also be described in the manuscript.

Author Response

Comment #1: The description of the electrospinning process needs to be further refined. In particular, how the pleated substrates were fixed on the drum to maintain its shape.

Response: We thank the reviewer for their suggestion. The electrospinning paper substrate was stretched and flattened with pleated lines during the electrospinning and fixed in the way as normal. After electrospinning, we release the stretch force to let the sheet spring back according to the folds we made before. This has been further clarified now in Section 2.4 of the manuscript.

Comment #2: The microstructure, filtration efficiency and pressure drop of craft tissue paper without electrospinning should also be described in the manuscript.

Response: We thank the reviewer for their suggestion. The SEM of tissue paper was added in SI for better comparison. The PFE and PD of the Tengujo tissue paper substrate was described in line 222 of the manuscript is 0.77 ± 0.13% and 2.1 ± 0.4 Pa.

Reviewer 3 Report

In this paper, a plant derived protein, zein was used to prepare a compostable air filter material on a craft paper-based substrate with high particle filtration efficiency by electrospinning. Some major points are needed to be revised before publication.

1- The introduction section of the MS should be improved. There are many research papers related to this study and should be cited. e.g. ACS Omega, 2023 8, 1453; Materials Science and Engineering B 279 (2022) 115675; Materials Science and Engineering B 264 (2021) 114953.

2- Table 1 is cited as "Table 2" within the text, which is correct, Table 1 or Table 2? 

In Table 1, the electrospinning parameters are reported. Both voltage and flow rate are changed at the same time. What is the reason for this change and how the change of experimentaÅŸ parameters affected the resultant fibers? This point should also be disccussed in details. 

3- The magnification of SEM images must be given. The scale of SEM images is the appropriate, 10 and 20 micrometers scale should be corrected according to the diameter of the fibers. It would be better to use smaller scale.

4- FTIR spectra could be given to identify the chemical structure of the filter materials

5- For mechanical testing, tensile strength, yield strength and/or modulus of elasticity values should be measured and the mechanical properties of the filter materials than can be compared.

6- Conclusion section should contain discussion of the results, briefly. Authors can compare the results of the prepared filter materials and report the best one. 

Author Response

Comment #1: The introduction section of the MS should be improved. There are many research papers related to this study and should be cited. e.g. ACS Omega, 2023 8, 1453; Materials Science and Engineering B 279 (2022) 115675; Materials Science and Engineering B 264 (2021) 114953.

Response: We thank the reviewer for their comment. We have made several changes to the introduction section to clarify our emphasis on developing biodegradable filtration materials for barrier face covering applications (face masks for bioaerosols). We carefully went through the references suggested and unfortunately could not find a way that they would add to the discussion here as they mostly seem to refer to non-biodegradable air filter materials that are not immediately relevant for this manuscript. Hence we have decided to leave them out.

Comment #2: Table 1 is cited as "Table 2" within the text, which is correct, Table 1 or Table 2? 

In Table 1, the electrospinning parameters are reported. Both voltage and flow rate are changed at the same time. What is the reason for this change and how the change of experimentaÅŸ parameters affected the resultant fibers? This point should also be disccussed in details. 

Response: We appreciate that the reviewer took note of our typographical error and have corrected it in the main content. For the electrospinning parameters, the viscosity was increased significantly with increasing concentration. As a result, both the voltage and flow rate were adjusted to get the optimal parameters for continuous fiber generation. The flow rate was increased to help push the higher viscosity solution through the tubing, and higher voltage was used to increase the electrical field strength to result in continuous nanofiber production

Comment #3: The magnification of SEM images must be given. The scale of SEM images is the appropriate, 10 and 20 micrometers scale should be corrected according to the diameter of the fibers. It would be better to use smaller scale.

Response: We appreciate the reviewer’s suggestion.  We have updated our SEM images to show higher resolution figures. The magnification of SEM images was 8000x for the concentration and the scale of the SEM images has been changed to 2 micrometers in the manuscript as shown in Figures 2 and 3.

Comment #4: FTIR spectra could be given to identify the chemical structure of the filter materials.

Response: We thank the reviewer for the suggestion. The FTIR spectra has been provided in the SI file to help identify the chemical structure of the filter materials.

Comment #5: For mechanical testing, tensile strength, yield strength and/or modulus of elasticity values should be measured and the mechanical properties of the filter materials than can be compared.

Response: We thank the reviewer for the comment. The mechanical robustness referred by the authors in this manuscript relates to the attachment of the electrospinning layer and substrate. For the crosslinked samples, the attachment was much more robust to handle the manual folding, which was common in the practical application, compared to the uncrosslinked samples. Tensile testing may not be suitable for this material as the electrospun layer is considerably weaker than the paper that it sits on and therefore the properties would be that of the backing paper predominantly.

Comment #6: Conclusion section should contain discussion of the results, briefly. Authors can compare the results of the prepared filter materials and report the best one. 

Response: We thank the reviewer for their suggestion. The values of various parameters were added and highlighted in the manuscript for better understanding of the readers.

Round 2

Reviewer 2 Report

All issues in the manuscript have been corrected and it could be accepted.

Reviewer 3 Report

The authors revised the MS as requested and most of the major issues are included. The paper is acceptable in the present form.